# miR-129a-3p Inhibits PEDV Replication by Targeting the EDA-Mediated NF-κB Pathway in IPEC-J2 Cells

**DOI:** 10.3390/ijms22158133

**Published:** 2021-07-29

**Authors:** Xiaoyi Qi, Yue Cao, Shenglong Wu, Zhengchang Wu, Wenbin Bao

**Affiliations:** 1College of Animal Science and Technology, Yangzhou University, Yangzhou 225000, China; xiaoyiqi0321@163.com (X.Q.); yuecao97@163.com (Y.C.); slwu@yzu.edu.cn (S.W.); zcwu@yzu.edu.cn (Z.W.); 2Joint International Research Laboratory of Agriculture and Agri-Product Safety, Yangzhou University, The Ministry of Education of China, Yangzhou 225000, China

**Keywords:** pigs, PEDV, miR-129a-3p, *EDA*, transcriptome sequencing

## Abstract

Previous studies have shown that microRNAs (miRNAs) are closely related to many viral infections. However, the molecular mechanism of how miRNAs regulate porcine epidemic diarrhea virus (PEDV) infection remains unclear. In this study, we first constructed a PEDV-infected IPEC-J2 cytopathic model to validate the relationship between miR-129a-3p expression levels and PEDV resistance. Secondly, we explored the effect of miR-129a-3p on PEDV infection by targeting the 3′UTR region of the ligand ectodysplasin (*EDA*) gene. Finally, transcriptome sequencing was used to analyze the downstream regulatory mechanism of *EDA*. The results showed that after 48 h of PEDV infection, IPEC-J2 cells showed obvious pathological changes, and miR-129a-3p expression was significantly downregulated (*p* < 0.01). Overexpression of miR-129a-3p mimics inhibited PEDV replication in IPEC-J2 cells; silencing endogenous miR-129a-3p can promote viral replication. A dual luciferase assay showed that miR-129a-3p could bind to the 3′UTR region of the *EDA* gene, which significantly reduced the expression level of *EDA* (*p* < 0.01). Functional verification showed that upregulation of *EDA* gene expression significantly promoted PEDV replication in IPEC-J2 cells. Overexpression of miR-129a-3p can activate the caspase activation and recruitment domain 11 (*CARD11*) mediated NF-κB pathway, thus inhibiting PEDV replication. The above results suggest that miR-129a-3p inhibits PEDV replication in IPEC-J2 cells by activating the NF-κB pathway by binding to the *EDA* 3′UTR region. Our results have laid the foundation for in-depth study of the mechanism of miR-129a-3p resistance and its application in porcine epidemic diarrhea disease-resistance breeding.

## 1. Introduction

Porcine epidemic diarrhea (PED) is a highly contagious intestinal disease caused by porcine epidemic diarrhea virus (PEDV) [1,2]. Pigs of all ages are affected, but especially suckling piglets, and the virus is mainly manifested in piglets as watery diarrhea, vomiting, dehydration, and loss of appetite, with a mortality rate as high as 90%, causing serious harm to the pig industry worldwide [3]. PEDV belongs to the coronavirus family; it is an α-coronavirus, which is a kind of enveloped single strand positive RNA virus [4]. At present, vaccination is an effective way to prevent and control PEDV infection, but the variation of PEDV strain makes the virulence of mutant strains increase. Some studies have found that the existing PEDV vaccine cannot produce effective immune protection against the mutant strains of PEDV [5,6]. Therefore, the most effective and fundamental way to control PEDV infection is to screen functional genes and effective molecular markers to improve the resistance of piglets to PEDV.

MicroRNAs (miRNAs) are endogenous non-coding single-stranded RNA molecules, approximately 18–23 nucleotides in length, that regulate the expression of target genes by binding to the 3′ untranslated region of target genes [7]. At present, studies have found that miRNAs play an important role in many biological processes, such as cell proliferation, differentiation, apoptosis, stress response, and inflammatory pathway regulation [8,9,10,11,12,13]. In addition, miRNA expression in host cells is significantly affected during virus infection. On the one hand, the host cell changes miRNA expression, which may be due to the virus escaping the antiviral immune response by changing the intracellular environment. On the other hand, the host cells may trigger antiviral defenses and affect virus replication [14,15]. More and more evidence shows that miRNA plays an important role in virus infection. miRNAs seem to have great potential as molecular targets for diagnosis and treatment of diseases. Recognition of the miRNA-mediated regulation mechanism of PEDV infection will help to identify new targets for anti-PEDV treatment. miR-129a-3p was initially found in porcine adipose tissue and milk exosomes [16,17]. Núñez et al. performed microRNA sequencing in the tonsils and mediastinal lymph nodes (MLN) of porcine circovirus (PCV2)-infected and uninfected pigs and found that miR-129a expression is upregulated in MLN and may be involved in pathways related to the immune system and processes related to PCV2 pathogenesis [18]. However, the mechanism of miR-129a-3p in piglets infected with PEDV is unclear. In order to further explore the mechanism of miR-129a in PEDV-infected pig intestinal epithelial cells, in this study, qPCR was used to verify the relationship between miR-129a-3p and PEDV infection. Secondly, bioinformatics analysis, a dual luciferase reporter system, and Western blot were used to verify the important target gene ligand ectodysplasin (*EDA*) of miR-129a-3p. The regulation of PEDV resistance in weaned piglets was further analyzed by means of target gene overexpression, indirect immunofluorescence, and TCID50. Finally, the downstream pathways regulated by target genes were analyzed by transcriptome sequencing. The aim was to reveal a new molecular mechanism by which miR-129a-3p regulates PEDV resistance and lay the foundation for in-depth study of the application of miR-129a-3p in breeding for disease resistance to porcine epidemic diarrhea.

## 2. Results

### 2.1. Effect of miR-129a-3p on PEDV Viral Replication

Pathological changes in PEDV-infected cells were observed under a microscope. The wild-type IPEC-J2 cells were full fusiform, and 24 h post-infection, cell shrinkage, elongation, and fusion were observed. Further, 48 h post-infection, the cells shrank into granules with partial cell necrosis. All cells shrank and deformed at 72 h (Figure 1A). We then used RT-qPCR to assess the differential expression of miR-129a-3p at various timepoints after PEDV infection. The results showed that the expression of miR-129a-3p was significantly downregulated at 24 h and 48 h (*p* < 0.01) (Figure 1B). miR-129a-3p mimics and inhibitor were transfected into IPEC-J2 cells, respectively, and the expression of intracellular miR-129a-3p was detected using RT-qPCR after 48 h. The results showed that the expression level of miR-129a-3p in the IPEC-J2 cells from the miR-129a-3p group was significantly higher than that from the NC group (*p* < 0.01), the miR-129a-3p inhibitor correspondingly showed significantly inhibited expression levels of miR-129a-3p (Figure 1C). This suggests that synthetic miR-129a-3p mimics and inhibitors can be used in subsequent experiments. RT-qPCR indicated that within 48 h of the infection, viral particle copies in the miR-129a-3p mimics group were significantly reduced (*p* < 0.01), and the virus particle copies in the miR-129a-3p inhibitor group were significantly increased (*p* < 0.05) (Figure 1D). TCID50 experiments showed that PEDV infected J2 cells had significantly fewer infected virions after transfection with miR-129a-3p mimics (*p* < 0.01), virions infected with miR-129a-3p inhibitor were efficiently increased (*p* < 0.01) (Figure 1E). Indirect immunofluorescence showed that PEDV particles could be observed in all four treatment groups; compared with the NC group, PEDV particles in the mimics group decreased significantly, and significantly more PEDV viral particles were detected in the inhibitor group compared with that of the inhibitor NC group (Figure 1F). Western blot was used to detect the expression of PEDV N protein (QNL15265.1), and it was found that the expression of PEDV N protein was significantly lower in the mimics group compared with that of the NC group (Figure 1G); PEDV N protein expression was increased in the inhibitor group compared with that of the inhibitor NC group (Figure 1H). In summary, miR-129a-3p inhibited the replication of PEDV in IPEC-J2 cells.

### 2.2. miR-129a-3p Targets EDA

miR-129a-3p was found to have binding sites with *EDA* by website prediction (Figure 2A). The expression of *EDA* was examined at different time points of PEDV infection, and the results showed that the *EDA* gene showed a highly significant upregulated expression at 24 h and 48 h (*p* < 0.01) (Figure 2B). To further explore the effect of miR-129a-3p on the expression of *EDA*, the expression levels of *EDA* were measured by RT-qPCR, and the results showed that *EDA* mRNA was significantly downregulated by miR-129a-3p mimics (*p* < 0.05), and the inhibitor significantly upregulated the expression of *EDA* (*p* < 0.01) (Figure 2C). Correlation analysis of the expression changes of miR-129a-3p and *EDA* genes at different time points of PEDV infection was performed by SPSS 24.0 software, the results showed that the expression level of miR-129a-3p and *EDA* showed a very significant negative correlation (*p* < 0.01) (Figure 2D). Simultaneously, we successfully constructed EDA-WT and EDA-MUT vectors (Appendix A). The EDA-WT and EDA-MUT vectors were transfected into IPEC-J2 cells, and luciferase activity was measured. The luciferase activity was significantly inhibited in miR-129a-3p mimics + EDA-WT compared with that of the NC + EDA-WT group (*p* < 0.01); miR-129a-3p mimics + EDA-MUT had no significant change in luciferase activity compared with that of NC + EDA-MUT (*p >* 0.05) (Figure 2E). The results of WB showed that the expression of EDA protein (XP_005657872.1) was decreased by miR-129a-3p mimics (Figure 2F) and miR-129a-3p inhibitor promoted EDA protein expression (Figure 2G). Therefore, miR-129a-3p was able to target *EDA* to downregulate its expression level.

### 2.3. Effect of the EDA Gene on PEDV Viral Replication

IPEC-J2 cells were transfected with *EDA* overexpression plasmids and fluorescence was visualized 24 h later, the results indicated that the *EDA* overexpression plasmid was successfully transfected into the cells (Figure 3A). Total RNA and protein were collected from cells after drug screening, and the expression levels of the target gene *EDA* were determined by RT-qPCR. The results showed that the *EDA* expression level of the EDA-OE group was extremely significantly upregulated compared with that of the control group (*p* < 0.01) (Figure 3B). The expression of EDA protein was detected by Western blot analysis, which showed that the EDA-OE group showed higher expression of EDA protein compared with that of the NC group (Figure 3C). Fluorescence RT-qPCR results of PEDV *M* gene showed that 24 h after PEDV infection of the cells, the expression of PEDV *M* gene in the EDA-OE group was significantly higher than that in the NC group (*p* < 0.01) (Figure 3D). The TCID50 experiment showed that the infected virus particles in the IPEC-J2 cells infected by PEDV in the EDA-OE group increased significantly (*p* < 0.01) (Figure 3E). The indirect immunofluorescence experiment found that PEDV virus particles in the EDA-OE group were significantly increased compared with that of the NC group (Figure 3F). The amount of PEDV N protein was determined by Western blot analysis, which revealed that the expression of PEDV N protein was significantly higher in the EDA-OE group compared with that of the NC group (Figure 3G). In summary, *EDA* overexpression promoted PEDV replication in IPEC-J2 cells.

### 2.4. Mechanism of Downstream Regulation of the EDA Gene

Transcripts from *EDA* overexpressing cell lines (EDA-OE), control cells (NC), IPEC-J2 cells, and PEDV infected control cells (PEDV-J2) were obtained by RNA seq. Adjusted *p* < 0.05 and |log2 fold change| > 2 were used as criteria to screen differentially expressed genes. The results showed that there were 214 differentially expressed genes between the two groups, including upregulation of 123 and downregulation of 91 in EDA-OE vs. NC (Appendix A). A total of 943 mRNAs were differentially expressed between PEDV-J2 and J2 groups, of which 598 were upregulated and 345 were downregulated (Appendix A). Heatmap plots of the filtered, differentially expressed genes showed similar gene expression patterns within groups and distinct gene expression patterns between groups (Figure 4A,B, Appendix A). Intersecting the differential genes between EDA-OE and NC and PEDV-J2, it was found that there are 83 differential genes in common between the two groups (Figure 4C, Appendix A). Gene expression patterns were similar within groups and significantly different between groups (Figure 4D). Using string (https://string-db.org, accessed on 27 April 2021) site prediction analysis of protein interactions of these differential genes revealed that most of the differential genes interacted with *IL8* (Appendix A). Differentially expressed gene (DEGs) are mainly involved in oxoacid metabolic processes, organic acid metabolic processes, and GO classification (Figure 4E). Cluster analysis was performed to screen the differentially expressed genes DEGs and found that they were mainly involved in 10 pathways; the most significant differences were in the phospholipase D signaling pathway, NF−κB signaling pathway, and arachidonic acid metabolism (Figure 4F).

Six genes were selected for RT-qPCR based on the differential fold and biological function of gene expression as a result of transcriptome sequencing, and the selected differential genes were *CARD11*, *OASL*, *PTGS2*, *CXCL8*, *IL16*, and *IL19*. The results show that the RNA-seq results of those six genes were consistent with the expression trends of the RT-qPCR results (Figure 4G,H). By examining the expression of these differential genes between the miR-129a-3p mimics group and the inhibitor group, we found that the expression changes of these six genes in the miR-129a-3p mimics group showed an opposite trend to that of the EDA-OE group by transcriptome sequencing. The expression changes of these six genes in the miR-129a-3p inhibitor group were consistent with the results of transcriptome sequencing of the EDA-OE group (Figure 4I). This indicates that miR-129a-3p can target *EDA* and then regulate these differential genes.

### 2.5. miR-129a-3p Activates the EDA-Mediated NF-κB Pathway

After adding NF-κB pathway inhibitor BAY 11-7028 to cells, WB detects the phosphorylation of p65; the results showed that the phosphorylation level of p65 in the BAY 11-7028 group was significantly lower than that in the NC group, indicating that the NF-κB pathway inhibitor had a significant effect (Figure 5A). RT-qPCR results revealed that miR-129a-3p mimics could significantly reduce the expression level of the PEDV *M* gene after transfection into cells (*p* < 0.01). Cells transfected with miR-129a-3p mimics and infected with PEDV were treated with BAY 11-7028, and we found that the expression level of the PEDV *M* gene was significantly increased after BAY 11-7028 treatment compared with that of the control (*p* < 0.01) (Figure 5B). The expression of *CARD11* was assessed by RT-qPCR, which showed that the expression level of *CARD11* was significantly increased in the mimics + PEDV compared with that of the NC + PEDV group; *CARD11* expression levels were significantly reduced in mimics + PEDV + BAY 11-7028 compared with that of mimics + PEDV (*p* < 0.01) (Figure 5C). WB results showed that the expression changes of the PEDV N protein and CARD11 protein were consistent with the quantitative results; the phosphorylation level of p65 was increased in the mimics + PEDV compared with that in the NC + PEDV group, p65 phosphorylation was decreased in mimics + PEDV + BAY 11-7028 compared with that of the mimics + PEDV (Figure 5D). The changes of cytokines *1L-12*, *IL-1β*, *IFN-α*, *IFN-β*, and *IFN-γ* were detected by RT-qPCR. The results showed that the levels of *1L-12*, *IL-1β*, *IFN-α*, *IFN-β*, and *IFN-γ* in the mimics + PEDV group were higher than those in the NC + PEDV group (*p* < 0.05) (Figure 5E).

## 3. Discussion

microRNAs (miRNAs) are post transcriptional gene regulators, which interact with the 3′ untranslated region (3′UTR) of target genes to fine-tune gene expression and protein synthesis and regulate cell signaling pathways. The target gene mRNA transcript contains a large number of microRNA response elements (MREs); microRNAs can play a role in combination. In addition, each microRNA may inhibit up to hundreds of transcripts. Therefore, it is estimated that microRNA regulates a large proportion of the transcriptome [19]. Studies have shown that virus infection can cause changes in miRNA expression, which plays an important role in regulating host genes and virus replication [20]. For example, changes in miR-324-5p expression in A549 cells can affect the replication of the H5N1 virus [21]. miR-27 can affect adenovirus infection by targeting the 3′UTR region of *SNAP25* and *TXN2* [22]. miR-22 can change the replication of PRRSV in cells [23]. However, the role of miRNA in PEDV infection remains unclear.

In this study, we found that the expression level of miR-129a-3p was closely related to the infectivity of PEDV. IPEC-J2 cells from different treatment groups were infected with PEDV to detect PEDV copy number, and PEDV infection was examined by indirect immunofluorescence experiments. The results showed that the replication ability of the miR-129a-3p mimics group was significantly lower than that of the control group. Compared with the control group, the replication ability of PEDV in the miR-129a-3p inhibitor group was significantly upregulated. At present, the quantitative detection method for infectious PEDV live virus still relies on the determination of viral plaque forming units (PFU) and median cell culture infectious dose (TCID50) [24]. In this study, the TCID50 detection method showed that compared with the control group, the number of virus particles in the miR-129a-3p mimics group was significantly reduced and the miR-129a-3p inhibitor group had an effective increase in infected virions. WB results showed that the protein level of PEDV in the miR-129a-3p mimics group was significantly lower than that in the control group, and the protein level of PEDV in the miR-129a-3p inhibitor group was significantly higher than that in the control group. Therefore, miR-129a-3p plays an important role in the process of PEDV infection. Upregulation of miR-129a-3p expression can improve the resistance of IPEC-J2 cells to PEDV.

EDA protein is a type II transmembrane protein in the TNF superfamily [25]. It has been found that the *EDA* gene severely affects the development of some organs and structures of the ectoderm, for example, the skin, hair, and nails [26]. However, the role of the *EDA* gene in PEDV-infected cells has not been reported. In this study, we found through website prediction that miR-129a-3p and *EDA* 3′UTR exist in the binding region. The expression level of the *EDA* gene is closely related to the infectivity of PEDV. miR-129a-3p mimics were subsequently transfected with inhibitor into IPEC-J2 cells and the results showed that *EDA* mRNA expression levels were significantly reduced in the miR-129a-3p mimics group compared with that of the control group and *EDA* mRNA expression was significantly increased in the miR-129a-3p inhibitor group. The expression of miR-129a-3p and *EDA* at different time points of PEDV infection were analyzed; the results showed that miR-129a-3p was negatively correlated with *EDA*. A dual luciferase report showed that miR-129a-3p decreased luciferase activity by interacting with *EDA* 3′UTR, initially demonstrating that *EDA* is a target gene of miR-129a-3p. WB results showed that the expression of the EDA protein in the miR-129a-3p mimics group was significantly lower than that in the control group and the expression of the EDA protein in the miR-129a-3p inhibitor group increased significantly. In conclusion, miR-129a-3p inhibits the replication of PEDV in IPEC-J2 cells by targeting the 3′UTR region of *EDA* to reduce the expression of *EDA*.

To further explore the role of *EDA* in PEDV-infected IPEC-J2 cells, in this experiment, we constructed an *EDA* gene-overexpressing IPEC-J2 cell line. The results showed that the viral replication ability was significantly upregulated and the intracellular PEDV viral particles were also increased in the EDA-OE control group compared with those in the control group. The expression of the PEDV protein was detected by WB, and the amount of PEDV protein was found to be significantly increased in the EDA-OE group. In summary, upregulation of *EDA* expression facilitates PEDV replication in IPEC-J2 cells.

The transcriptome consists of mRNA, tRNA, rRNA, and noncoding RNA molecules; mRNA is an intermediate molecule between the DNA sequence and functional proteins. In gene expression studies, measuring the levels of different mRNAs in cells by RNA-seq technology is a common means of gaining insight into the biological activity of cells or tissues [27]. In this experiment, RNA-seq technology was used to obtain transcripts from *EDA* overexpressing cell lines, control cells (NC), porcine small intestinal epithelial cells (IPEC-J2), and PEDV-infested IPEC-J2 cells (PEDV-J2). Functional enrichment analysis of differential genes was performed, and six important candidate genes were screened in combination with biological functions. *CARD11*, a member of the membrane-bound guanylate kinase family, plays an important role in extracellular signaling mediated by T cell receptors and B cell receptors. It can bind to B-cell lymphoma 10, associated lymphoid tissue lymphoma translocator protein 1, and tumor necrosis factor receptor-related factor 6 to stimulate downstream pathways [28,29,30]. *PTGS2* is an inducible prostaglandin enzyme involved in inflammatory PG biosynthesis processes, and it is a key enzyme in inflammatory PG biosynthesis [31]. *OASL*, a member of the OAS family, is an interferon-inducible antiviral protein that plays an important role in innate immune processes, it is also involved in biological processes such as apoptosis and growth and differentiation [32,33,34]. Some studies have found that the expression level of *PTGS2* in the uterine tissue of sows induced by E. *coli* is significantly increased, and the expression of *PTGS2* in the uterus and glands is directly related to the intensity of organ inflammation, which further shows that *PTGS2* is very important for PG secretion in inflammatory organs [35]. *CXCL8* (*IL8*) is a novel inflammatory factor that plays an important role in acute inflammation by recruiting inflammatory cells to induce fibrotic processes [36]. *IL16* is a cytokine produced by CD8 + T cells that is able to induce CD4 + T cells, monocytes, as well as eosinophil chemotaxis and plays an important role in the immune process of biological organisms [37]. It has been found that *IL16* can inhibit mixed lymphocyte reaction and replication of immunodeficiency virus-1 (HIV-1) [38]. *IL19*, a member of the *IL10* family, mediates apoptosis by inducing the production of *IL6* and *TNF-α* by monocytes [39]. *IL19* plays an important role in host defense mechanisms against bacteria as well as induces immune responses [40]. These genes are differentially expressed between different groups and may be associated with immune responses against PEDV in pigs. In this study, we hypothesized that upregulation of *EDA* gene expression levels induces differential expression of candidate genes during PEDV infection, inhibits immune-related signaling pathways, and ultimately leads to enhanced toxic effects of PEDV on intestinal epithelial cells.

In this study, we found that miR-129a-3p mimics could reduce the expression levels of PEDV mRNA and protein after transfection into cells. Cells transfected with miR-129a-3p mimics and infected with PEDV were treated with BAY11-7028, and the results revealed that the expression levels of PEDV mRNA and protein were increased after BAY11-7028 treatment compared with those of the control group. Some studies have reported that activation of NF-κB can inhibit the replication of PEDV [41]. Combined with the results of this experiment, it is speculated that miR-129a-3p may inhibit the replication of PEDV in IPEC-J2 cells by activating the NF-κB pathway. Some studies have found that the *CARD11* gene can be involved in regulating the NF-κB pathway [42], it can mediate factor-specific activation of NF-κB through T cell receptor complexes [43]. In this study, we examined the expression of *CARD11* mRNA and protein in different treatment groups, and we found that miR-129a-3p mimics could upregulate the expression levels of *CARD11* mRNA and protein after transfection of cells; cells transfected with miR-129a-3p mimics and infected with PEDV were treated with BAY11-7028, and the results revealed that the expression levels of *CARD11* mRNA and protein after BAY11-7028 treatment were significantly reduced compared with those of the control group. These results indicate that miR-129a-3p mimics may activate the NF-κB pathway by altering *CARD11* expression. *p65* and *p50* are the most widespread and most active heterodimers of NF-κB; however, only *p65* has a transcriptionally active regulatory domain, which contains phosphorylation sites that can regulate the function of NF-κB, which in turn affects the expression of downstream genes [44]. To further investigate whether miR-129a-3p is able to activate the NF-κB pathway, we examined the phosphorylation of *p65* in different treatment groups, and we found that the phosphorylation level of *p65* was increased in mimics + PEDV compared with that of the NC + PEDV group, and it was decreased in mimics + PEDV + BAY11-7028 compared with that of mimics + PEDV. The above results suggest that miR-129a-3p inhibits PEDV replication in IPEC-J2 cells by activating the NF-κB pathway by binding to the *EDA* 3′UTR region.

In conclusion, the results of the present study revealed that miR-129a-3p inhibited PEDV replication in IPEC-J2 cells by activating the NF-κB pathway by binding the *EDA* 3′UTR region (Figure 6), which uncovered the new molecular mechanism of how miR-129a-3p regulates PEDV resistance. Our findings lay the foundation for in-depth study of miR-129a-3p in pig epidemic diarrhea disease-resistance breeding, and the method used in this study can provide guidance for identifying functional genes and regulators of other virus infections in further studies.

## 4. Materials and Methods

### 4.1. Cell Culture and PEDV Infection

Vero cells, 293 cells, and IPEC-J2 cells provided by the University of Pennsylvania (Philadelphia, PA, USA) were cultured in DMEM complete medium containing 10% FBS (Grand Island life science Co., Ltd., New York, NY, USA). IPEC-J2 cells were inoculated into a 12-well plate until 70% confluence. They were then treated with PEDV-CV777 viral (National Type Culture Collection, Wuhan, China) solution (MOI = 0.1) for 24 h, 48 h, and 72 h. Pathological changes in the cells during each timepoint were observed under a microscope. Cells were collected at different time points for RT-qPCR analysis.

### 4.2. Plasmid Transfection

IPEC-J2 cells were seeded in 12-well plates for 24 h, followed by transfection of plasmid for 24 h using Lipofectamine 3000 (Invitrogen Co., Ltd., Carlsbad, CA, USA) followed by infection with PEDV at an MOI of 0.1. After 36 h or 48 h, RNA was collected for quantitative analysis. Total RNA was extracted and PEDV copy number was calculated based on the standard curve equation of PEDV-CV777 previously established by our research group: y = −3.3354lg(x) + 37.832.

### 4.3. Dual Luciferase-Reporter Assays

PCR amplification primers containing binding sites were designed according to the target region where miR-129a-3p recognizes and binds *EDA* (F: AATAAAAGATCCTTTATTAAGCTTTGGCACAAGAAGCAGCTGTA, R: TCATAGGCCGGCATAGACGCGTCCCTCCCTCTAGCCTCAGTT, italicized representatives of digestion sites). The recombinant vector was the *EDA* wild-type vector (pMIR-Report-EDA-WT) containing the miR-129a-3p binding site. The *EDA* mutant vector containing the miR-129a-3p binding site was kindly provided by Genecreate (Wuhan Genecreate Biotechnology Co., Ltd., Wuhan, China) (pMIR-Report-EDA-MUT). The plasmid was transfected into IPEC-J2 cells using Lipofectamine 3000, with 3 replicates per treatment group: set pMIR-Report-EDA-WT + NC, pMIR-Report-EDA-WT + miR-129a-3p mimics, pMIR-Report-EDA-MUT + NC, pMIR-Report-EDA-MUT + miR-129a-3p mimics group, set control group (pMIR-Report Luciferase), and blank cell group. The cells were collected 48 h post-transfection, according to the Dual Luciferase^®^ Reporter Assay System (Nanjing Novezan Biotechnology Co., Ltd., Nanjing, China). The firefly luciferase (Ff) and renilla luciferase (Rn) activities were measured using a multifunctional fluorescence detector. The ratio of Ff/Rn was used as the luciferase activity value.

### 4.4. TCID50 Analysis

Viral titers were determined by the 50% tissue culture infective dose (TCID50) method. The Vero cells were resuspended and plated into 96-well plates, to achieve 60% cell density, and were seeded with PEDV with a dilution of six gradients in each well. Eight replicates were done for each treatment and eight blank controls were left. The experimental results were recorded each day.

### 4.5. Immunofluorescence Assays

IPEC-J2 cells were seeded in a 12-well plate at a density of 5 × 10^5^ cells/mL until 70% confluence. After 24 h of PEDV infection, the cells were washed thrice with PBS and fixed in 4% formaldehyde (Nanjing Novezan Biotechnology Co., Ltd., Nanjing, China) at room temperature for 60 min. Then, 0.5% Triton X-100 (Nanjing Novezan Biotechnology Co., Ltd., Nanjing, China) was added, and the mixture was incubated for 15 min, 5% BSA (Nanjing Novezan Biotechnology Co., Ltd., Nanjing, China) was subsequently added and the reaction was allowed to proceed for 2 h. The supernatant was discarded, followed by the addition of a primary antibody (Veterinary Medical Research & Development Co., Ltd., Washington, DC, USA) (1:500) and secondary detection antibody (Abcam Co., Ltd., Cambridge, UK) (anti-mouse IgG, 1:200). The cells were then stained with DAPI (Abcam Co., Ltd., Cambridge, UK) (1:800) and observed under a fluorescence microscope.

### 4.6. EDA Overexpression Cell Lines

Flag-EDA was synthesized by Genecreate (Wuhan Genecreate Biotechnology Co., Ltd., Wuhan, China) and cloned into pCDH-CMV-MCS-EF1-CopGFP+Puro (pCDH-GFP) using the *Nhe I* and *Kpn I* restriction sites. Transfection with the *EDA* overexpression plasmid and empty vector was performed until the cell density reached about 70%. After 24 h of transfection, 5 μg/mL puromycin was used for drug screening; after the positive cell expression was stable, cellular RNA and protein were extracted for verification of overexpression efficiency.

### 4.7. RT-qPCR Analysis

Total RNA was extracted from cells using an RNA extraction kit (Nanjing Novezan Biotechnology Co., Ltd., Nanjing, China). Total RNA purity and concentration were assessed using 2% formaldehyde agarose gel electrophoresis and NanoDrop 1000. RNA samples were stored at −80 °C. RNA was reversely transcribed into cDNA using a reverse transcription kit (Nanjing Novezan Biotechnology Co., Ltd., Nanjing, China). RT-qPCR was conducted using a kit of AceQ qPCR SYBR Green Master Mix (Nanjing Novezan Biotechnology Co., Ltd., Nanjing, China). Three replicates were set for each sample and the CT values were averaged. Melting curves were analyzed after finishing to judge the specificity of the PCR primers. The fluorescent quantitative primers were designed by Primer 5.0 software, and *GAPDH* was used as the internal reference gene (Appendix A).

### 4.8. Western Blot Analysis

Cells were lysed with protein lysate buffer for 20 min on ice, followed by centrifugation at 12,000× *g* for 10 min at 4 °C Protein samples were added to 5× loading buffer, boiled for 10 min, subjected to sodium dodecyl sulfate polyacrylamide gel electrophoresis, and transferred to polyvinylidene fluoride membranes, which were blocked with 5% nonfat milk and probed using the primary monoclonal antibody (PEDV antibody, Veterinary Medical Research & Development Co., Ltd., Washington, DC, USA; Phospho-NF-κB p65 antibody, Cell Signaling Co., Ltd., Danvers, MA, USA; EDA antibody, Santa Cruz Co., Ltd., Dallas, TX, USA; CARD11 antibody, AffinitY Co., Ltd., Shanghai, China; Hsp90 antibody, abcam Co., Ltd., Cambridge, MA, USA). Horseradish peroxidase-conjugated secondary antibodies (Nanjing Novezan Biotechnology Co., Ltd., Nanjing, China) were used to detect the primary antibodies, and proteins were visualized by ECL.

### 4.9. RNA-seq and Computational Analysis

Briefly, total RNA was extracted from the *EDA* gene overexpression cell line (EDA-OE, n = 4), porcine intestinal epithelial cells (IPEC-J2, n = 4), and PEDV-infected porcine intestinal epithelial cells (PEDV-J2, n = 4). All RNA samples were used to synthesize double-stranded cDNA and sequencing was performed at the Novogene Bioinformatics Institute (Beijing, China) on an Illumina Hiseq 2500 platform (Illumina Inc., San Diego, CA, USA). Transcriptome analysis was performed according to the manufacturer’s instructions, followed by data analysis as described previously [45]. The analysis process included six steps: data quality control, reference genome alignment, quantitative gene expression analysis, RNA-Seq correlation analysis, difference significance analysis, and functional enrichment.

### 4.10. Statistical Analysis

Results of the RT-qPCR were analyzed by the 2^−ΔΔCt^ method [46]. SPSS 18.0 software was used to analyze the data. Student’s *t* test was used for all statistical analyses. All data are presented as the mean ± SD of three independent experiments. The * denotes a significant difference with *p* < 0.05, while the ** denotes a significant difference with *p* < 0.01.

## Figures and Tables

**Figure 1 ijms-22-08133-f001:**
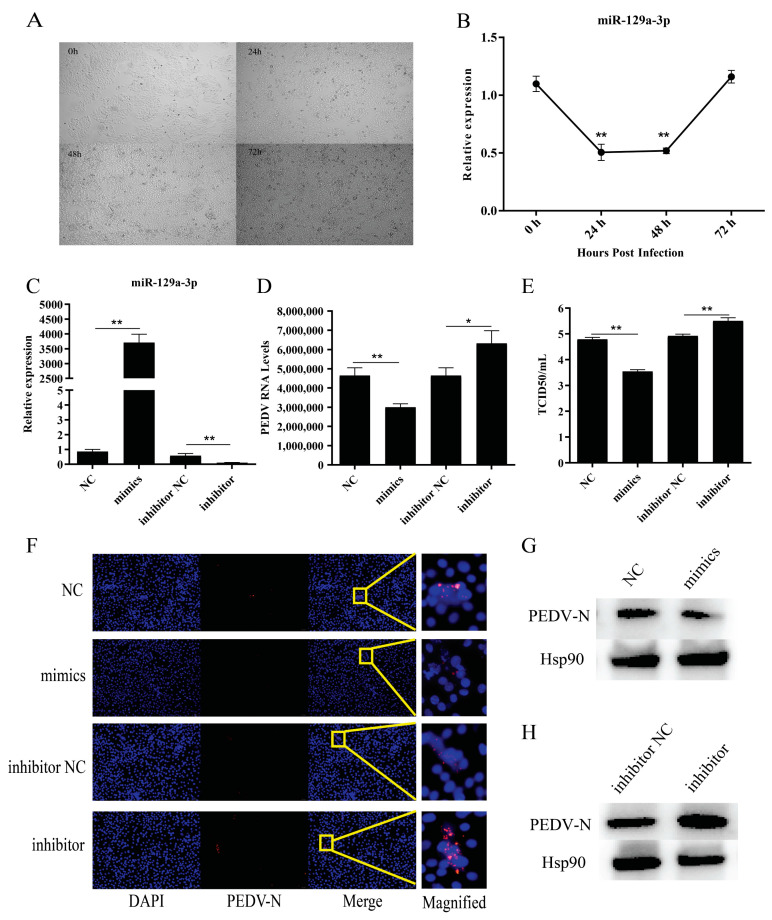
Effect of miR-129a-3p on PEDV viral replication. (**A**) Cell morphology at different time points in IPEC-J2 cells infected with PEDV. (**B**) Analyzing miR-129a-3p expression after PEDV infection. (**C**) Validation of miR-129a-3p mimics and inhibitor efficiency. (**D**) Differential analysis of PEDV copy number in cells infected with PEDV with different treatments. (**E**) Analysis of TCID50 results of cells infected with different treatment groups by PEDV. (**F**) Effects of different treatments on PEDV replication. (**G**) Expression levels of PEDV protein after PEDV infection of cells treated with miR-129a-3p mimics. (**H**) Expression levels of PEDV protein after PEDV infection of cells treated with miR-129a-3p inhibitor. The * denotes a significant difference with *p* < 0.05, while the ** denotes a significant difference with *p* < 0.01.

**Figure 2 ijms-22-08133-f002:**
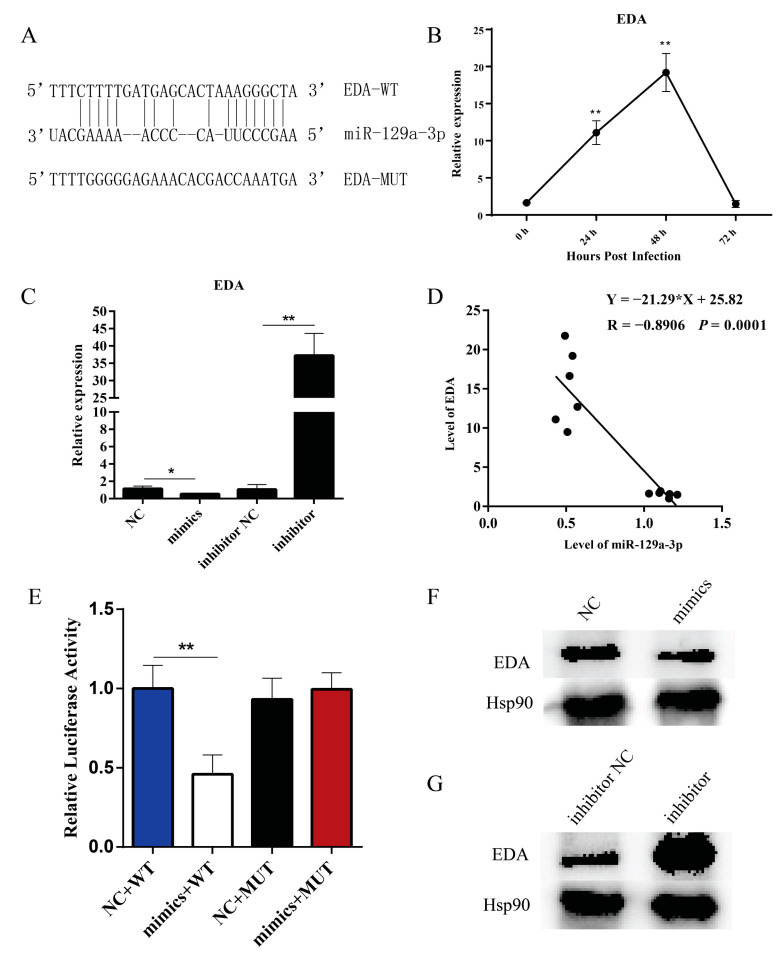
miR-129a-3p targets *EDA*. (**A**) miR-129a-3p and *EDA* complementary sequence and *EDA* mutant sequence. (**B**) Expression changes of the *EDA* gene after IPEC-J2 infection by PEDV. (**C**) Expression levels of the *EDA* gene in different treatment groups. (**D**) Expression correlation analysis between miR-129a-3p and *EDA*. (**E**) Dual luciferase assay was performed to verify the targeting relationship between miR-129a-3p and *EDA*. (**F**) Expression levels of intracellular EDA protein after miR-129a-3p mimics treatment. (**G**) Expression levels of intracellular EDA protein after miR-129a-3p inhibitor treatment. The * denotes a significant difference with *p* < 0.05, while the ** denotes a significant difference with *p* < 0.01.

**Figure 3 ijms-22-08133-f003:**
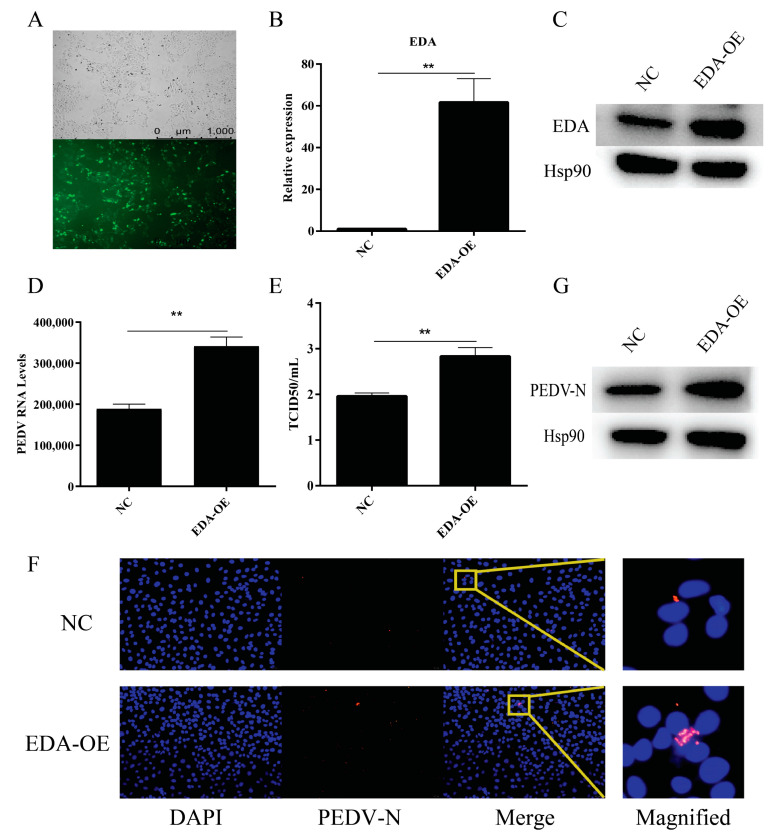
Effect of the *EDA* gene on PEDV viral replication. (**A**) The transfection efficiency of the *EDA* overexpression plasmid; the upper panel shows the corresponding cells in the absence of fluorescence exposure, showing the overall density of the cells. (**B**) Expression levels of *EDA* gene in different treatment groups. (**C**) Expression levels of EDA protein in the different treatment groups. (**D**) Differential analysis of PEDV copy number in cells infected with PEDV with different treatments. (**E**) Analysis of TCID50 results of cells infected with different treatment groups by PEDV. (**F**) Effects of different treatments on PEDV replication. (**G**) Expression levels of EDA protein in different treatment groups. The ** denotes a significant difference with *p* < 0.01.

**Figure 4 ijms-22-08133-f004:**
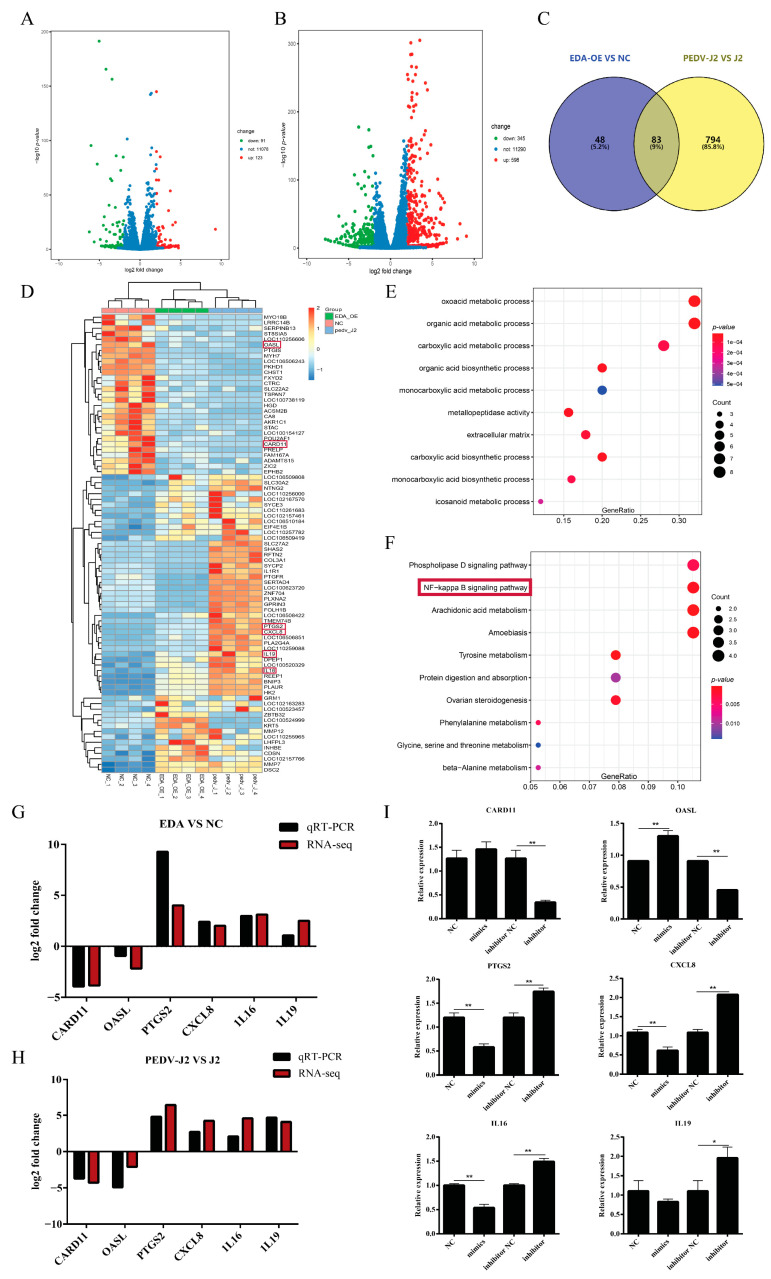
Mechanism of downstream regulation of the *EDA* gene. (**A**) Volcano plot of differential genes between the two groups in EDA-OE vs. NC. (**B**) Volcano plot of differential genes between the two groups for PEDV-J2 versus J2. (**C**) Common differential genes between EDA-OE versus NC and PEDV-J2 versus J2 groups. (**D**) Cluster analysis graph of differential gene samples between two groups. (**E**) GO enrichment histograms. (**F**) Differential gene KEGG enrichment scatter plot. (**G**) Transcriptome sequencing validation of EDA-OE vs. NC. (**H**) Transcriptome sequencing validation of PEDV-J2 vs. J2. (**I**) Differential gene expression changes in different treatment groups. The * denotes a significant difference with *p* < 0.05, while the ** denotes a significant difference with *p* < 0.01.

**Figure 5 ijms-22-08133-f005:**
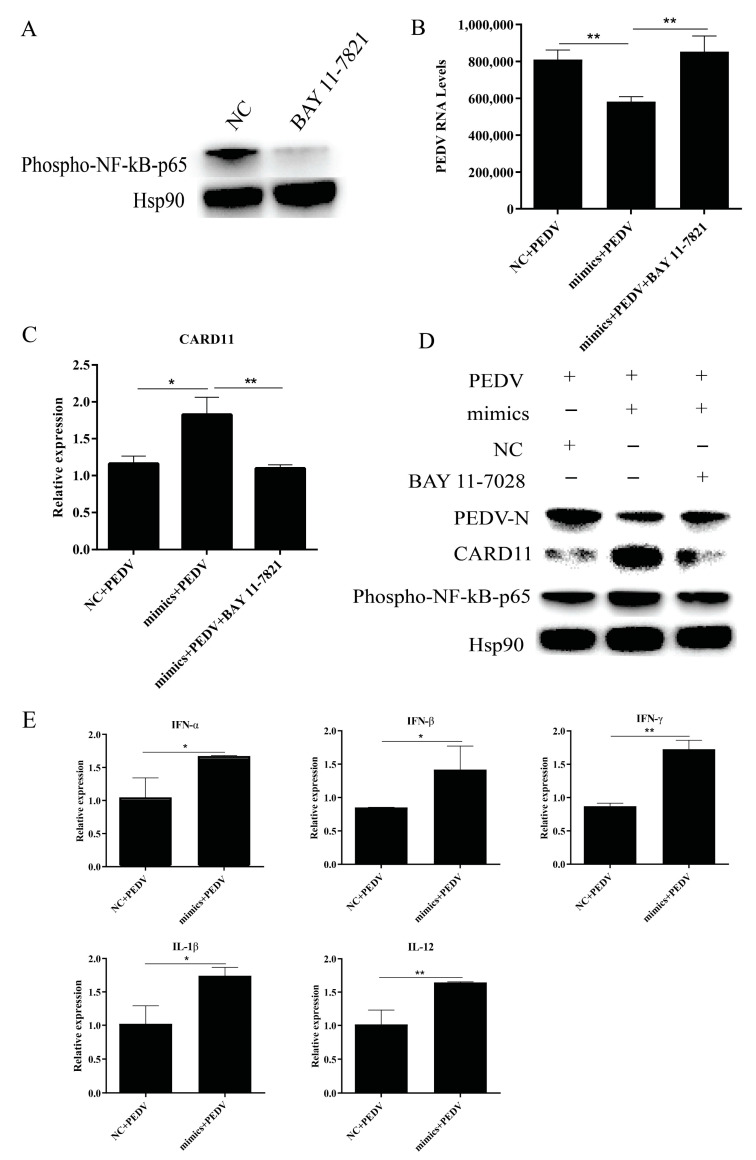
miR-129a-3p activates *EDA*-mediated NF-κB pathway. (**A**) WB results after BAY 11-7028 treatment. (**B**) Changes of PEDV *M* gene expression in different treatment groups. (**C**) Changes of *CARD11* gene expression in different treatment groups. (**D**) WB results of PEDV-N, CARD11, and Pp65 proteins in different treatment groups. (**E**) Changes of *1L-12*, *IL-1β*, *IFN-α*, *IFN-β,* and *IFN-γ* gene expression in different treatment groups. The * denotes a significant difference with *p* < 0.05, while the ** denotes a significant difference with *p* < 0.01.

**Figure 6 ijms-22-08133-f006:**
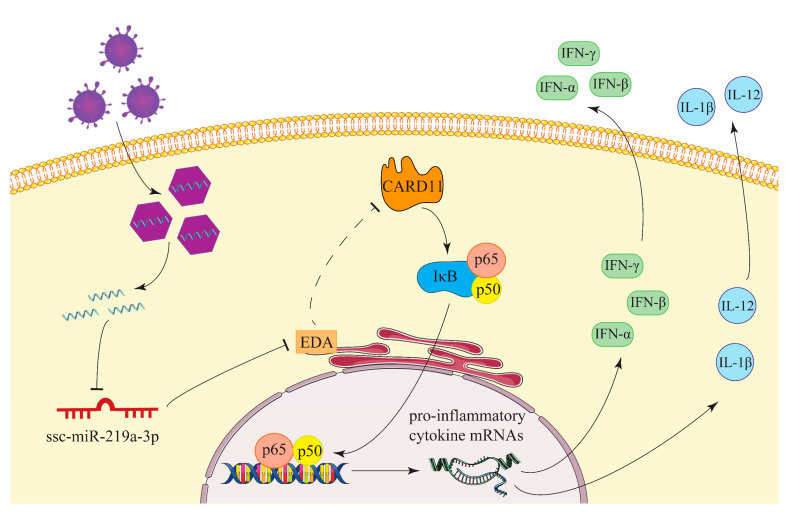
miR-129a-3p inhibits PEDV replication by targeting the EDA-mediated NF-κB pathway in IPEC-J2 cells. Ligand ectodysplasin, EDA; Caspase activation and recruitment domain 11, CARD11; NF-κB IκB, IκB; NF-κB p65, p65; NF-κB p50, p50; Interferon-alpha, IFN-α; Interferon-beta, IFN-β; Interferon-gamma, IFN-γ; Interleukin-12, IL-12; Interleukin-1beta, IL-1β.

## Data Availability

The data presented in this study are available on request from the corresponding author.

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
