# Peer review of "miR-129a-3p Inhibits PEDV Replication by Targeting the EDA-Mediated NF-κB Pathway in IPEC-J2 Cells"

_ijms, 2021, doi:10.3390/ijms22158133_

Round 1

Reviewer 1 Report

Thank you very much for the opportunity to review. I recommend this manuscript for publication.

General comment

The study reported miR-129a-3p inhibits PEDV replication by targeting EDA-me-
diated NF-κB pathway in IPEC-J2 cells

In general, the paper was well written and understandable. I recommend this manuscript for publication. 

Specific comments

1. The conclusion section should be expanded, for example:

- for other viral diseases, the author's methods whether can also be used

- how this method can be applied to other infectious disease eradication?

Author Response

Dear editors and reviewers,

Thank you for your careful revision and comments on our manuscript. We made a correction and hope to get approval. The revised part is marked on the paper with different colors. The main corrections in the paper and the responses to the reviewer’s questions are detailed in the attachment.

Reviewer 2 Report

Qi et al described the miR-129a-3p inhibits PEDV replication by targeting EDA-mediated NF-kB pathway in IPEC-J2 cells. This is an interesting research finding about PEDV. This manuscript is suitable for publish in International Journal of Molecular Sciences after minor revision.

Comments:

  1. What is EDA? Abbreviation should be defined the first time when it is used.
  2. Figure 1A, authors mentioned PEDV infected cells show shrink and deform at 72 h, I would like to know does authors observed multinucleated giant cells, the typical early CPE of coronavirus infection.
  3. Figure 1D, 1E, the scale interval of y-axis is 1 rather than 2.
  4. Figure 2B, the title of x-axis, “Hours Post Infection”
  5. Figure 2D, the title of x- and y-axis, “level of miR-129a-3p” and “level of EDA”, respectively.
  6. Page 11, results, 2.6. I would like suggested move Figure 6 into discussion section.
  7. Page 11-12, line 249-268 and line 283-312. this is discussion section, however, authors do not necessary to descript the results again. I would like recommend authors re-write the discussion, Line 249-312.

Author Response

(The authors gave the same response as above.)
